# A validity study to consult on a protocol of a home hazard management program for falls prevention among community dwelling stroke survivors

Husna Ahmad Ainuddin[1,2], Muhammad Hibatullah Romli[1,3]*, Mazatulfazura S. F. Salim[1], Tengku Aizan Hamid[3], Lynette Mackenzie[4]

1 Department of Rehabilitation Medicine, Faculty of Medicine and Health Sciences, Universiti Putra Malaysia, Serdang, Selangor, Malaysia, 2 Centre of Occupational Therapy Studies, Faculty of Medicine and Health Sciences, Universiti Teknologi MARA, Puncak Alam, Selangor, Malaysia, 3 Malaysian Research Institute on Ageing, Universiti Putra Malaysia, Serdang, Malaysia, 4 Discipline of Occupational Therapy, Faculty of Health Sciences, The University of Sydney, Lidcombe, NSW, Australia

* hibatullah.romli@gmail.com

## Abstract

### Objective

A fall after a stroke is common but the consequences can be devastating not only for the stroke survivors, but also for caregivers, healthcare, and the society. However, research on falls prevention among the stroke population are limited, particularly on home hazards assessment and home modifications, demanding for a study to be conducted. The aim of the study is to validate the protocol and content of a home hazard management program guided by the Person-Environment-Occupation (PEO) Model for falls prevention among community dwelling stroke survivors.

### Method

Researchers developed their own questionnaire for content validation which consist of 23 items that covers two domains, namely justification for telehealth home hazard management practice and the protocol's overall methodology. Occupational therapists with at least one year of experience in conducting a home hazard assessment were consulted for the content validation of a two-group clinical controlled trial protocol utilizing a home hazard assessment, home modifications and education over the usual care. Written consent was obtained prior to the study. The occupational therapists were given a Google Form link to review the protocol and intervention based on the questionnaire and rated each item using a four-point Likert scale for relevance and feasibility. Open-ended feedback was also recorded on the google form. Content Validity Index (CVI), Modified Kappa Index and Cronbach's Alpha was calculated for the content validity and reliability analysis.

### Results

A total of sixteen occupational therapists participated in the study. 43.7% of participants had a master's degree, 93.7% worked in the government sector and 56.2% had six years and

**Data Availability Statement:** Data cannot be shared publicly because of confidentiality. Data are

available from the Ethics Committee (contact via jkeupm@upm.edu.my) for researchers who meet the criteria for access to confidential data.

**Funding:** M.H.R Grant Number: FRGS/1/2020/SS0/UPM/02/27 Name of funder: Ministry of Higher Education, Malaysia. URL: https://jpt.mohe.gov.my/portal/index.php/ms/penyelidikan/mygrants The funders had no role in study design, data collection and analysis, decision to publish, or preparation of the manuscript.

**Competing interests:** The authors have declared that no competing interests exist.

more experience on conducting home hazard assessments. Content validity of the protocol is satisfactory for relevancy and feasibility (CVI = 0.84, ranging from 0.5 to 1.00), and for the reliability ($\alpha$ = 0.94 (relevance) and $\alpha$ = 0.97 (feasibility), respectively. The Modified Kappa ranged from 0.38 to 1.00 for all items. Feedback was also received regarding the design and procedure of the study protocol which included participant's selection criteria, sample size, equipment provided, cost, location, and care for the participants during the intervention.

## Conclusions

Introducing a home hazard management program to prevent falls among the stroke population is viewed relevant and feasible. Practical suggestions from the consultation panel were adopted, and minor adjustments were required to strengthen the protocol's overall methodology. This study established a rigorous and robust experimental protocol for future undertaking.

## Introduction

The prevalence of falls among the stroke population is well documented worldwide [1]. Falls is one of the most common complications after a stroke in ten Asian countries including Malaysia [2]. For the stroke individual, the consequences of a fall may involve physical complications [3, 4], psychosocial issues [5, 6], decline in functioning and even mortality. Falls in stroke may increase the burden of caregivers and healthcare and involve high costs to society [7]. Other significant risk factors of post-stroke falls include a history of previous falls [8–10], reduced balance [10, 11], lower functional status [12, 13], fear of falling [14, 15], depressive symptoms [16], physical and sensory impairment [14], spatial neglect [17] and environmental safety hazards [18]. The increasing incidence of falls pose a challenge for rehabilitation regardless of whether the fall leads to serious or non-serious injuries [19]. Therefore, fall prevention is of importance for stroke survivors, their caregivers, healthcare practitioners and community.

A recent systematic review [19] examined the effectiveness of interventions for preventing falls after a stroke. The interventions included physical exercise, multifactorial interventions, active repeated transcranial direct current stimulation. Interventions on environment and assistive technology are available in the review [19], however, interventions for home hazards were not comprehensive. The studies investigated on pre-discharge assessments comparison, and simple aids such as type of glasses and walking aids showing an insignificant difference on falls rate. Another review by Ainuddin et al. [20] also revealed that studies on falls and stroke were one dimension and neglected several crucial aspects of falls risk and prevention particularly environmental factors. Majority of the studies focused on intervening the physical impairments of stroke in general and considered environmental management as challenging due to clinician's and client's barriers [20, 21].

The body of knowledge on falls is greatest in older population and this can be applied to stroke population. Stroke and falls are a pertinent issue in older population and is frequently addressed simultaneously [22]. Falls risk factors [23], intervention and prevention studies [24, 25] among the elderly population have been conducted extensively. However, one significant factor identified from falls in older population literature but was limitedly examined in stroke research was environmental factors [19, 20]. Home assessment and modifications in the

homes of older people were reported to be effective in reducing the rate and risk of falls either as a single intervention [26] or part of a multifactorial intervention [27–29]. Home hazards management has been investigated extensively with older people and also with other age-associated disorders such as Alzheimer's, dementia and with caregivers, and found beneficial to improve falls prevention, functionality, self-care and independency, physical health and well-being, social participation, economy, and caregiving [30–32]. Several guidelines identified from Australia [33], the US [34] and Europe [35–37] recommend interventions to reduce environmental hazards for falls prevention. The focus rests on the premise that falls among older people can be reduced by assessing and modification of the environment and this can be achieved through two key mechanisms which are minimizing known falls hazards and changing how a person interacts with their surroundings [38].

Hazards are assessed via observations of the home and how the client functions in their home. This concept is known as person-environment fit [39]. Most often, a home assessment and modification are conducted by occupational therapists via a home visit. Studies reported that home hazards are significantly associated with falls [40, 41] however, a study by Romli et al. [42] indicated that occupational therapists found it difficult to perform home visits, evaluations and implement modification because of clinical duties, resources and differences in clients' experiences and views. While healthcare professionals regarded home visits as essential to avoid falls and injuries, clients believed that these visits are unnecessary as it increased their financial costs and the home modifications suggested were not pleasing in terms of aesthetic value. Furthermore, some clients resented anyone disturbing their home or rearranging their furniture. Some believed that their falls were due to poor attention to their surroundings and not related to existing environmental hazards [43]. These barriers hindered effective home assessments as a strategy to prevent falls within the community.

In addition to home hazard assessments, environmental interventions also include increasing awareness of risks of falls and problem solving with the older person to find solutions to improve function, independence and/or safety (e.g., modifying a shower to improve access) and the use of assistive technology to assist independence (e.g., provision of mobility aids, grab rails, and personal alarms) [26]. Gitlin [44] defined home environmental modifications as a vast array of strategies that include structural renovation, assistive devices, placement of visual cues and memory aids, and rearrangement or removal of furniture and dangerous household items as well as the simplification of tasks. As the interior space of home may become a place of safety and security [45], ergonomics in floor designs and appropriate posture techniques [46] are also advised as part of the intervention as these techniques could minimize barriers and increase supportive features to facilitate participation in activities of daily living and leisure within the home [45]. Limitation in methodologies which include low risk populations [28], the lack of adherence to the program and differences in the trial populations may also have influenced the outcomes [47], as well as adopting a checklist as opposed to using a functional environmental evaluation [48] which could have decreased the sensitivity in detecting home hazards.

Telehealth approach is recently gaining demand due to its increase access to care [49], cost and time efficient while maintaining patient satisfaction [50]. Over the past two decades, literature supporting remote home modification interventions has continued to grow [51]. A systematic review found that clinical outcomes were comparable for remote and on-site home visits [52]. Feasibility studies have demonstrated that restricted technology, providing two-way audio and video communication, enables occupational therapists to deliver successful remote home environment modifications [51, 53]. However, limitations in the methodologies have demanded a more rigorous study design to be applied. Adopting a framework as the basis of the methodology could enhance implementation of the study.

### Theoretical framework

Fall prevention guidelines emphasize the need for careful evaluation of an individual's risks and deficits [54] and consideration of the interaction between multiple risk factors is needed [34]. The Person–Environment–Occupation (PEO) model is a useful framework to apply to fall risk evaluation and align with the person-environment fit concept [39, 55, 56]. The model's key assumption is that the person, environment, and occupation interact continuously across time and space in ways that increase or decrease their congruence: the closer the fit, the greater the overlap of roles, routines, and tasks [56]. Roles, routines, and tasks in the context of fall prevention focuses on reducing fall risk and enhancing clients' confidence in their ability to engage in valued activities without falling [54]. This study which is built upon the PEO model aims to improve the roles, tasks, and routines of community stroke survivors through an individually designed home hazard management program to prevent falls and increase participants confidence to conduct daily activities without falling. Fig 1 illustrates the PEO model.

As this will be the first study to examine falls and home hazards among the stroke population, a pilot quasi-experimental trial would be the most appropriate study design as such practice has not been explored in Malaysia. This is to ensure that the instruments used in the study are appropriate and interventions developed are safe and beneficial for the participants that can derived a desired outcome for research [57]. Initially, the intervention was created based on a literature review. Therefore, this study aimed to collect opinions from experienced occupational therapists for the content validation of the developed program to be implemented among community dwelling stroke survivors.

## Materials and methods

### The protocol

From the gap identified, a protocol is planned to investigate the effectiveness of home hazards management via telehealth with stroke survivors on several outcomes. The protocol will apply a pilot two-group non-randomized controlled trial. Non-randomized controlled trial is chosen as it allows a comparison with the availability of a control group and provides optimal blinding effort with a minimal cost. Stroke survivors are eligible to be considered for the study by certain subjective criteria and objective screening such as using the Modified Rankin Scale [58] and 6-Cognitive Impairment Test [59]. Stroke survivors will be enrolled either in an experimental or control group according to the hospital of recruitment. The experiment group utilizes home hazards management program which includes an online home hazard assessment and modification as well as education and training on falls prevention strategies, ergonomics, work simplification and energy conservation within the home based on the Home Falls and Accident Screening Tool [60] finding. Meanwhile, the control group will only receive standard care. The outcomes were measured using prospective falls calendar [61] for three months for falls rate, Falls Efficacy Scale–International [62] for fear of falling, Stroke Impact Scale 3.0 [63] and Canadian Occupational Performance Measure (COPM) [64] for functional performance, Short Form-12 [65] quality of life and wellbeing, and Zarit's Caregiver Interview [66] to assess caregiver burden. All of the outcome measures have been translated into the local languages at least in Bahasa Melayu as it is the national language and additionally some in Mandarin and Tamil, and all translated versions of the outcome measures have been validated. The detail of the protocol is presented in S1 Table.

### Study design of the consultation

The consultation members were recruited in a cross-sectional study design to validate the home hazard management program among community dwelling stroke survivors. Typically,

recruiting an experienced panel is organized to generate consensus and propose a solution for an issue with limited information or new challenges [67]. The method is appropriate as feedback is provided by experienced professionals. Therefore, the outcome is considered trustworthy and reliable to be implemented in practice. This study has received ethical approval from the Medical Research Ethical Committee, Ministry of Health Malaysia (NMRR-20-501-52933) and Universiti Putra Malaysia Ethics Committee (JKEUPM-2021-166).

## Participants and recruitment

Members of the consultation were recruited via voluntary participation. The google form link were blasted through different occupational therapy Whatsapp groups and those who were

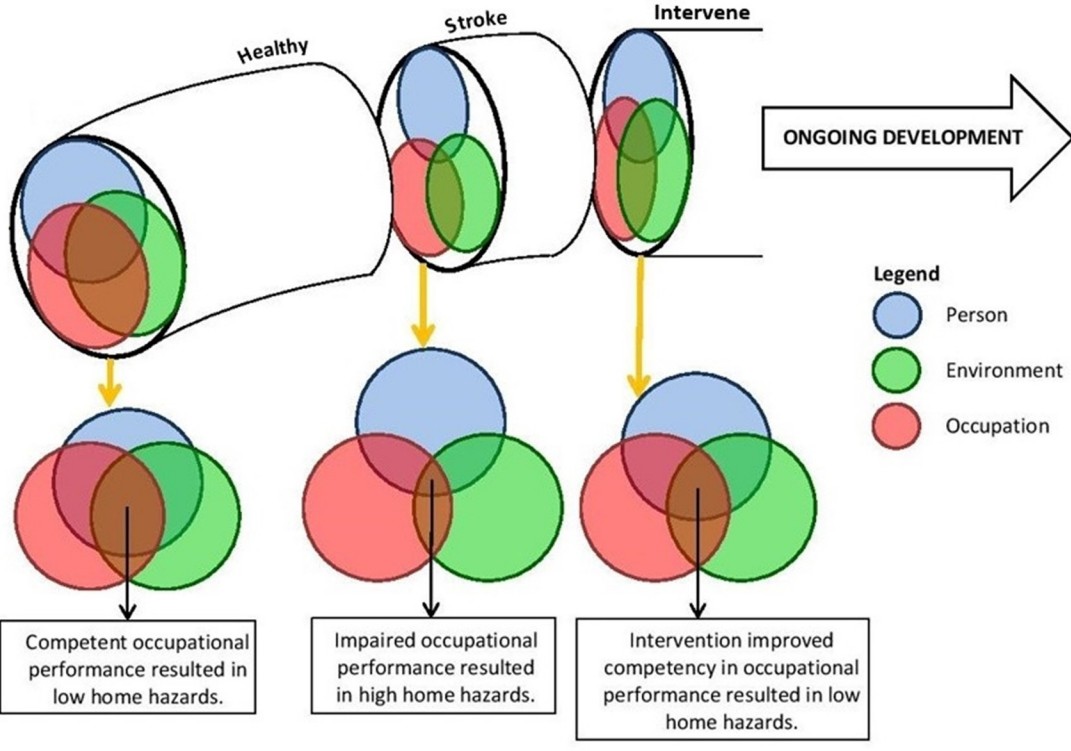

| Lifespan | Healthy (no disability) | Stroke (with disabilities) | Stroke (intervene) |
|---|---|---|---|
| Person | • Good balance<br>• Good eyesight<br>• Intact cognitive<br>• Fast reaction | • Poor balance<br>• Good eyesight<br>• Impaired cognitive<br>• Delayed reaction<br>• Incontinence | • Poor balance<br>• Good eyesight<br>• Reminder for toilet before bed<br>• Delayed reaction<br>• Schedule timing for toilet |
| Environment | • Ample size toilet (without rail)<br>• Manual light switch | • Ample size toilet (without rail)<br>• Manual light switch | • Ample size toilet (handrail installed)<br>• Manual light switch<br>• Additional automatic motion-sensor light installed |
| Occupation | • Going to toilet | • Rushing to go to toilet | • Going to toilet |
| Occupational performance | *No hazards risk for falling* | *Hazardous risk for falling* | *Less hazards risk for falling* |

**Fig 1. PEO model interaction depicting hazards occurrence.**

interested to participate and fulfilled the inclusion criteria filled up the form. Inclusion criteria included i) occupational therapists, and ii) have at least one year experience conducting home visits and hazard assessments. Consent was obtained prior to their participation in the study. Only occupational therapists are selected to participate as previous studies indicated home modifications are typically conducted by occupational therapists and yield better outcomes [27].

### Questionnaire for consultation panel

From the literature, there is no standard guideline for designing and evaluating validity of a study protocol [68]. Thus, the researchers developed their own questionnaire to test the validity of the intervention which consist of 23 questions constructing two domains. The first domain is on justification for telehealth home hazards management practice which consist of 10 questions developed from the findings of a qualitative study [21]. The second domain having 13 questions investigated the protocol's methodology on title, study design, location, methods, materials, intervention, outcome measures, cost, and duration based on the suggestion by Rajadhyaksha [69]. Members were asked to rate the protocol's relevance and feasibility on a four-point Likert scale (where 1 = not feasible/ relevant, 2 = quite relevant/ feasible, 3 = relevant/feasible, 4 = very relevant/feasible). Open-ended feedback was also recorded from the google form.

### Procedure

Each consultation member was invited to review the protocol. Members were given a Google Form link to review the protocol and intervention based on the aforementioned questionnaire.

### Analysis

The Content Validity Index (CVI) in particular, the Item-CVI (I-CVI) is reported as suitable analysis for individual responses [70]. I-CVI is calculated as the number of experts giving a rating of "quite relevant" or 'highly relevant for each item and divided by the total number of experts. Values range from 0 to 1; where I-CVI $\geq$ 0.80 indicates the item is relevant and <0.8 requires further attention and amendment [70]. Additionally, a Modified Kappa Index was computed to estimate the I-CVI [71, 72]. The Modified Kappa (k) is an index of agreement among experts that indicates beyond chance that the item is relevant, clear, or another characteristic of interest [72]. The standards recommended by Cicchetti and Sparrow [73] were used to interpret k in which evaluation criteria for kappa is the values above 0.74, between 0.60 and 0.74, and the ones between 0.40 and 0.59 are considered as excellent, good, and fair, respectively. Meanwhile, the internal consistency for the protocol was analyzed using Cronbach's Alpha in which a value of 0.80 and above was deemed acceptable [74]. Statistical Package for Social Sciences (SPSS) version 22 was used for data analysis. The open-ended responses were synthesized narratively for the qualitative feedback.

## Results

A total of 16 occupational therapists reviewed and rated the proposed intervention. The demographic data of the consultants is presented in Table 1.

### Protocol validity

For ease of analysis, the Likert scale response were collapsed from a 4-point to a 2-point scale [75]. The response of '1 = not relevant/feasible' and '2 = quite relevant/feasible' was considered as 'poor' response. A score of '3 = relevant/feasible' and '4 = very relevant/feasible' was

**Table 1. Demographics of consultation members.**

|  | n | % |
|---|---|---|
| **Gender** |  |  |
| Male | 6 | 37.5 |
| Female | 10 | 62.5 |
| **Race** |  |  |
| Malay | 12 | 75.0 |
| Chinese | 2 | 12.5 |
| Others | 2 | 12.5 |
| **Education** |  |  |
| Diploma | 4 | 25.0 |
| Bachelor | 5 | 31.3 |
| Master | 7 | 43.7 |
| **Work Sector** |  |  |
| Government | 15 | 93.7 |
| Private | 1 | 6.3 |
| **Work Setting** |  |  |
| Clinical | 11 | 68.7 |
| Community | 4 | 25.0 |
| Education | 1 | 6.3 |
| **Work Experience** |  |  |
| 1–3 years | 3 | 18.8 |
| 4–5 years | 0 | 0.0 |
| 6–10 years | 6 | 37.5 |
| More than 10 years | 7 | 43.7 |
| **Experience Conducting Home Hazard Assessment** |  |  |
| 1–3 years | 4 | 25.0 |
| 4–5 years | 3 | 18.8 |
| 6–10 years | 6 | 37.4 |
| More than 10 years | 3 | 18.8 |

considered as 'good'. Table 2 details out the content validity result of the protocol. The I-CVI findings were found to be between 0.50 and 1.00, which indicates certain aspects requires further attention, justification, or modification.

For justification on telehealth home hazards management need domain, items which had I-CVI value of below than 0.80 are home visit limitations, use of telehealth as an alternate solution to home visits, aesthetic value, and perceived cost of home modifications. While on the protocol methodology, the less satisfactory ratings are only on study location. The internal consistency for relevance of all the items had a Cronbach Alpha value of $\alpha = 0.94$ while for feasibility, the Cronbach Alpha value was $\alpha = 0.97$. This indicates that the responses are strongly consistent for each question from all panel members.

## Narrative notes

All written feedback received were reviewed and analyzed. Several of the suggestions were addressed to strengthen the intervention program and ensure the future study's robustness.

One participant commented whether 'near falls' was counted as a fall. In addition, the comments and suggestions included justification on why to exclude aphasic stroke patients, age of the respondents, to include wheelchair bound stroke patients and to exclude patients with

**Table 2. Content validity of the protocol.**

| Item | Relevance/ Feasibility | | |
|---|---|---|---|
| | I-CVI* | P_c ** | k*** |
| **Title** | 0.88 | 0.002 | 0.88 |
| **Justification:** | | | |
| a. Falls is common after a stroke | 1.00 | 1.53 | 1.00 |
| b. Falls result in injuries, limit ADLs and social participation and increase caregivers' burden. | 1.00 | 1.53 | 1.00 |
| c. Home visit is time consuming and not cost effective | 0.50 | 0.20 | 0.38 |
| d. Telehealth could provide alternative solutions to home visit and assessment. | 0.63 | 0.12 | 0.57 |
| e. Home modification can prevent falls after stroke. | 0.94 | 0.000 | 0.94 |
| f. Home hazard assessment and home modification should be a routine in stroke rehabilitation management. | 1.00 | 0.000 | 1.00 |
| g. Stroke clients do not appreciate the aesthetic value of home modifications | 0.50 | 0.20 | 0.38 |
| h. Stroke client perceived home modification is expensive. | 0.75 | 0.03 | 0.74 |
| i. There are many cost-effective items for home modifications. | 0.75 | 0.03 | 0.74 |
| j. There are many aesthetic home modification items available | 0.88 | 0.002 | 0.87 |
| **Study protocol** (may refer to S1 Table) | | | |
| Study Objective | 0.94 | 0.000 | 0.94 |
| **Study Design** (including randomization and blinding) | 0.88 | 0.002 | 0.87 |
| **Participants** | | | |
| Inclusion Criteria | 0.88 | 0.002 | 0.87 |
| Exclusion Criteria | 0.88 | 0.002 | 0.87 |
| Recruitment | 0.88 | 0.002 | 0.87 |
| Sample Size | 0.81 | 0.009 | 0.81 |
| Location | 0.75 | 0.03 | 0.74 |
| **Data Collection** | | | |
| Outcome Measures | 0.88 | 0.002 | 0.87 |
| Procedure | 0.88 | 0.002 | 0.87 |
| **Home Modification Items and Assistive Devices** | 0.88 | 0.002 | 0.87 |
| **Cost** | 0.94 | 0.000 | 0.94 |
| **Duration of Study** | 0.94 | 0.000 | 0.94 |

Note: I-CVI: item-level content validity index, **pc: probability of a chance occurrence, k***: modified kappa

cognitive impairment. Furthermore, equipment for mobility aids were suggested to be provided for all study respondents. On sample size, consideration was noted on adding a 20% margin for dropouts during the study duration. Another occupational therapist raised the issue on ethical concerns on the control group of not being provided with any home hazard management during the study duration.

## Discussion

This study provides valuable information regarding the strength of the intervention program. French et al. [76] reported that one potential reason for unsuccessful execution of an intervention is due to lack of feasibility testing. Due to a lack of testing, an intervention may be inapplicable to practice, does not meet the demands of practitioners and patients, and is not practical for real-world implementation [77, 78]. Feasibility studies allowed researchers to discover any practical issues and adjust the research procedure while maintaining scientific accuracy and validity [79, 80]. As a result, this study was utilized to determine the key components needed

to create a quasi-experimental study, as well as to define the intervention protocol in detail for practice. Furthermore, this study demonstrates that protocols can be altered according to suggestions and be improved.

Implementing falls prevention programs for the stroke population should be a priority as the prevalence of falls after a stroke is still high and the consequence is greater [8, 81]. The proposed protocol was viewed as acceptable and feasible. This could be due to the fact that preventing falls in stroke requires intervention and one proven method is via home hazard management [27]. Although a home hazard assessment and modification are a common practice in the developed countries, limited resources hinder its implementation in Malaysia [21].

A home hazard assessment and home modification are often conducted on site and in-person as the gold standard. This study indicates the panel considers conventional home visit has its merit over the telehealth alternative. This reflects a previous study where therapists prefer doing home visits as they are familiar with such practice, although with constrains available [42]. However, evidence shows a reduce trend for home visits practice [82, 83] and the COVID-19 pandemic has made patients unable to access post-stroke rehabilitation services [84], including postponed and limited appointments, and limited physical contact. Telehealth technology needs to be considered as an alternative to overcome these challenges [85]. Telehealth is not a common practice in Malaysia and many healthcare professionals are not familiar with its implementation and uncertain with its effectiveness and applicability. However internationally, more telehealth practices are adopted as there is numerous recent supporting evidence are made available [84, 86, 87]. Therefore, more explanation is required to be given on this aspect when the actual study is conducted in the future.

Aesthetics are commonly left out when considering for home modifications. A study by Struckmeyer et al. [88] found that there was lack of available assessment tools that specifically target aesthetics as measuring the aesthetics could be a subjective task [88]. Furthermore, researchers may not have considered aesthetic appraisal to be of a priority to include in assessments of persons with disabilities' residential environments to date [88] as the nature of home modification aesthetics are sometimes unavoidable and rather focus on the practicality and benefits over the adornment. Education and discussion with clients on agreement on which aids they prefer or desire to purchase prior to the modification could facilitate aesthetic and functional appreciation of the equipment.

As for location, the selected hospitals are in distance with each other where the at-tended participants are mostly from the surrounding community. This may provide a natural blinding effort where the participants from both groups will have little chance to meet with each other which could prevent cross-communication. Three hospitals in the Klang Valley are conveniently selected through discussions among the research team. Klang Valley is a metropolitan area in Malaysia and is chosen due to the availability of facilities, supply and services related to home modifications to allow for prompt action to be performed. The selection of the hospitals is based on the criteria that the hospitals are a large-scale hospital, considered as the main hospital in the locality, receive and manage a large number of stroke cases that can supply a sustainable number of stroke survivors, and feasible for the researchers' surveillance. Although there is a concern on this aspect from the panel, however it is decided to maintain the current decision as the data indicates that this is not a major issue.

The panel considered the protocol's sample size as small. The opinion has its basis to ensure for a high-quality study, however, considering the aspect of this protocol is a pilot study, difficulty of recruiting suitable sample and conducting the intervention for home modification study as illustrated in previous literature [27], and constrained on resources in terms of financial and manpower, makes the suggested sample size for this protocol as reasonable. Hence, the sample size is considered optimal as suggested by Julious et al. [89] of between 12 and 18 participants per group

for a pilot study. Therefore, considering the feedback from the panel, the future protocol will consider the upper border of 18 participants per group rather than the current one which is considered the lower border of the sample size. Also, there was a query on the criteria of the samples, however, it is decided to maintain the original criteria in the protocol to ensure homogeneity [90] and minimise error in the outcome. The panel's concern is understandable from the perspective of practitioners which consider providing the widest possible service [91].

Misperception that home modification is expensive was similar with current literature [42, 43]. Weeks et al. [92] suggested that home modification can involves as simple as basic home safety which requires minimal or no cost (e.g., removing unsecured loose mat), simple home modification (e.g., providing adaptive equipment), and major modification (e.g., requires structural change). A report revealed that the most significant reasons respondents did not change their home or did not modify it to their desire were (a) their in-capacity to do it themselves and (b) their inability to afford the modifications [93]. However, a systematic review suggested home modification as the most cost-effective method for falls prevention intervention [94] and proper evaluation conducted by an expert such as an occupational therapist may prevent these issues as unnecessary or inadequate modifications were avoided as a result [92]. Hence, basic modification which includes handrails for indoor/outdoor stairs, grab rails for bathrooms, outside lighting, contrast edges for steps, non-slip bathmats and surfaces can be effective to reduce falls injury [95]. In our protocol, although the perception for the protocol's budget for home modification is relatively low, however, the required equipment and cost had been calculated systematically in which it was for the maximum clients' needs for any basic modifications and not considering for major modifications. Thus, the researchers have included contingency expenses and revised the budget for equipment purchase to RM 700.00 from the previous RM 625.00, considering the feedback from the panel.

There was a concern about the control group not receiving any modification. Thus, to ensure both ethical purpose and the integrity of the study outcome, the control participants will receive modification after they completed the 12 weeks of the study duration, if there is a need. Proper walking aids and mobility education will be considered as part of the recommended equipment following the feedback from panel. Overall, the proposed experimental trial study will help to evaluate the potential effectiveness of home assessment and modification in our practice and detect potential trends between current falls research populations. If successful, these approaches have the potential to be implement-ed in stroke healthcare services, should be able to minimize the needs of falls related health service and improve the quality of life of aging stroke survivors living in the community. For future studies, it is recommended that the protocol be reviewed qualitatively via focus group discussions of other healthcare professionals that are involved in falls preventions of stroke survivors as well as professional contractors or architects that have experience in providing home modifications for special populations.

## Conclusions

The content and protocol of the experimental trial study has been established and results from this consultation enhances the overall comprehensiveness and strength of the study. Introducing a home hazard management program to prevent falls among the aging stroke population is viewed relevant and feasible.

## Supporting information

**S1 Checklist. STROBE statement—checklist of items that should be included in reports of *cross-sectional studies*.**
(DOCX)

**S1 Table. Protocol of the quasi-experimental study.**
(DOCX)

## Acknowledgments

The authors would like to thank all occupational therapists involved for the review of this protocol.

## Author Contributions

**Conceptualization:** Husna Ahmad Ainuddin, Muhammad Hibatullah Romli.

**Formal analysis:** Husna Ahmad Ainuddin.

**Funding acquisition:** Muhammad Hibatullah Romli.

**Investigation:** Husna Ahmad Ainuddin.

**Methodology:** Husna Ahmad Ainuddin, Muhammad Hibatullah Romli.

**Resources:** Mazatulfazura S. F. Salim, Tengku Aizan Hamid, Lynette Mackenzie.

**Supervision:** Muhammad Hibatullah Romli.

**Writing – original draft:** Husna Ahmad Ainuddin.

**Writing – review & editing:** Muhammad Hibatullah Romli, Mazatulfazura S. F. Salim, Tengku Aizan Hamid, Lynette Mackenzie.

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
