## [Decision Letter · Decision Letter 0]

26 Aug 2022

PONE-D-22-17675Consultation on the relevance and feasibility of a home hazard management program for falls prevention among community dwelling stroke survivorsPLOS ONE

Dear Dr. Romli,

Thank you for submitting your manuscript to PLOS ONE. After careful consideration, we feel that it has merit but does not fully meet PLOS ONE’s publication criteria as it currently stands. Therefore, we invite you to submit a revised version of the manuscript that addresses the points raised during the review process.

We look forward to receiving your revised manuscript.

Kind regards,

Walid Kamal Abdelbasset, Ph.D.

Academic Editor

PLOS ONE

Journal Requirements:

“ASA was supported by a DFG Fellowship Grant (SA 3716-1-1) for the children studies.”

5. Please ensure that you refer to Figure 6 in your text as, if accepted, production will need this reference to link the reader to the figure.

7. PLOS requires an ORCID iD for the corresponding author in Editorial Manager on papers submitted after December 6th, 2016. Please ensure that you have an ORCID iD and that it is validated in Editorial Manager. To do this, go to ‘Update my Information’ (in the upper left-hand corner of the main menu), and click on the Fetch/Validate link next to the ORCID field. This will take you to the ORCID site and allow you to create a new iD or authenticate a pre-existing iD in Editorial Manager. Please see the following video for instructions on linking an ORCID iD to your Editorial Manager account: https://www.youtube.com/watch?v=_xcclfuvtxQ

Reviewers' comments:

Reviewer's Responses to Questions

**Comments to the Author**

1. Is the manuscript technically sound, and do the data support the conclusions?

Reviewer #1: No

2. Has the statistical analysis been performed appropriately and rigorously? 

Reviewer #1: Yes

3. Have the authors made all data underlying the findings in their manuscript fully available?

Reviewer #1: Yes

4. Is the manuscript presented in an intelligible fashion and written in standard English?

Reviewer #1: Yes

5. Review Comments to the Author

Reviewer #1: The manuscript entitled ‘Consultation on the relevance and feasibility of a home hazard management program for falls prevention among community-dwelling stroke survivors’ with the aim to validate the protocol and content of a home hazard management program guided by the Person-Environment-Occupation (PEO) Model for fall prevention among community-dwelling stroke survivors.

The manuscript requires improvement based on the following comments.

Introduction

Line 172-195, the protocol section in the introduction is to be placed in the Materials and Methods section.

Line 151, 207, 347, 361, there were numerous typographical errors e.g. ap-plied, vali-idate, at-tended, pi-lot.

Line 179, the non-random method used to be stated e.g. how the subjects were assigned to the groups.

Line 183, to state clearly the control group receives usual care only.

Line 194, is this study, feasibility or pilot study?

Materials and methods

It is not clear whether the content of all the tools is in English or local version. Since Malaysia consist of various ethnicity, will this protocol be made available in the local language if the current protocol is in the English version?

The actual protocol/content is to be included in the manuscript as appendices/supplementary.

The statistical software used in the analysis to be stated.

The rationale of having occupational therapists in the study evaluation and not involving other healthcare professionals such as occupational/rehabilitation/community/public health/geriatric doctors, hospital management staff etc or even caregivers is to be clearly stated.

Focus Group Discussion (FGD) could be employed in the study.

More samples could be recruited to strengthen the protocol especially with regards to validation aspect.

Table 1, the sample size could be calculated (if it is a pilot study) and the study design to be stated.

Results

Line 273, the Cronbach alpha values to be presented.

6. PLOS authors have the option to publish the peer review history of their article (what does this mean?). If published, this will include your full peer review and any attached files.

Reviewer #1: No

---

## [Author Response · Author response to Decision Letter 0]

28 Oct 2022

Review Comments to the Author

The manuscript requires improvement based on the following comments.

Comment: Line 172-195, the protocol section in the introduction is to be placed in the Materials and Methods section. 

Response: Protocol is placed in the Materials and Methods section (page 9, line 184)

Comment: Line 151, 207, 347, 361, there were numerous typographical errors e.g. applied, validate, attended, pilot. 

Response: Typographical errors were rectified 

Comment: Line 179, the non-random method used to be stated e.g. how the subjects were assigned to the groups.

Response: Stroke survivors will be enrolled either in an experimental or control group according to the hospital of recruitment (page 9, line 191)

Comment: Line 183, to state clearly the control group receives usual care only. 

Response: Rectified as suggested "Meanwhile, the control group will only receive standard care (page 9, line 196)"

Comment: Line 194, is this study, feasibility or pilot study? 

Response: The protocol is a feasibility study and pilot quasi-experimental trial. 

As this will be the first study to examine falls and home hazards among the stroke population, a feasibility study and pilot quasi-experimental trial would be the most appropriate study design as such practice has not been explored in Malaysia (page 8, line 175)

Comment: It is not clear whether the content of all the tools is in English or local version. Since Malaysia consist of various ethnicity, will this protocol be made available in the local language if the current protocol is in the English version? 

Response: A statement has been added "All of the outcome measures have been translated into the local languages at least in Bahasa Melayu as it is the national language and additionally some in Mandarin and Tamil, and all translated versions of the outcome measures have been validated."

Comment: The actual protocol/content is to be included in the manuscript as appendices/supplementary. Response: Protocol is as supplementary file

Comment: The statistical software used in the analysis to be stated.

Response: A statement has been added "Statistical Package for Social Sciences (SPSS) version 22 was used for data analysis (page 12, line 256)"

Comment: The rationale of having occupational therapists in the study evaluation and not involving other healthcare professionals such as occupational/rehabilitation/community/public health/geriatric doctors, hospital management staff etc or even caregivers is to be clearly stated.

Response: Only occupational therapists are selected to participate as previous studies indicated home assessments and modifications are typically conducted by occupational therapists and yield better outcomes [27] (page 10, line 222)

Comment: Focus Group Discussion (FGD) could be employed in the study.

Response: A statement of recommendation has been added "For future studies, it is recommended that the protocol be reviewed qualitatively via focus group discussions of other healthcare professionals that are involved in falls preventions of stroke survivors as well as professional contractors or architects that have experience in providing home modifications for special populations (page 19, line 407)"

Comment: More samples could be recruited to strengthen the protocol especially with regards to validation aspect.

Response: We just manage to get an extra 4 respondents were recruited to review the protocol. A total of 16 occupational therapists reviewed and rated the proposed intervention (page 12, line 261)

Comment: Table 1, the sample size could be calculated (if it is a pilot study) and the study design to be stated. Response: We decided for the sample size to be calculated after the pilot study. Therefore, we based our sample size on the suggestion from previous literature "Hence, the sample size is considered optimal as suggested by Julious et al. [89] of between 12 and 18 participants per group for a pilot study (page 18, line 370)"

Comment: Line 273, the Cronbach alpha values to be presented. 

Response: A statement has been added "The internal consistency for relevance of all the items had a Cronbach Alpha value of α=0.94 while for feasibility, the Cronbach Alpha value was α=0.97 (page 14, line 292)"

---

## [Decision Letter · Decision Letter 1]

14 Nov 2022

PONE-D-22-17675R1Consultation on the relevance and feasibility of a home hazard management program for falls prevention among community dwelling stroke survivorsPLOS ONE

Dear Dr. Romli,

Thank you for submitting your manuscript to PLOS ONE. After careful consideration, we feel that it has merit but does not fully meet PLOS ONE’s publication criteria as it currently stands. Therefore, we invite you to submit a revised version of the manuscript that addresses the points raised during the review process.

We look forward to receiving your revised manuscript.

Kind regards,

Walid Kamal Abdelbasset, Ph.D.

Academic Editor

PLOS ONE

Journal Requirements:

Reviewers' comments:

Reviewer's Responses to Questions

**Comments to the Author**

1. If the authors have adequately addressed your comments raised in a previous round of review and you feel that this manuscript is now acceptable for publication, you may indicate that here to bypass the “Comments to the Author” section, enter your conflict of interest statement in the “Confidential to Editor” section, and submit your "Accept" recommendation.

Reviewer #1: All comments have been addressed

Reviewer #2: All comments have been addressed

2. Is the manuscript technically sound, and do the data support the conclusions?

Reviewer #1: Partly

Reviewer #2: Yes

3. Has the statistical analysis been performed appropriately and rigorously? 

Reviewer #1: Yes

Reviewer #2: Yes

4. Have the authors made all data underlying the findings in their manuscript fully available?

Reviewer #1: Yes

Reviewer #2: Yes

5. Is the manuscript presented in an intelligible fashion and written in standard English?

Reviewer #1: Yes

Reviewer #2: Yes

6. Review Comments to the Author

Reviewer #1: (No Response)

Reviewer #2: Thanks for your efforts in editing the previous comments. Only the title of your manuscript should include the type of the study

7. PLOS authors have the option to publish the peer review history of their article (what does this mean?). If published, this will include your full peer review and any attached files.

Reviewer #1: No

Reviewer #2: No

---

## [Author Response · Author response to Decision Letter 1]

30 Nov 2022

Comment: Please review your reference list to ensure that it is complete and correct. If you have cited papers that have been retracted, please include the rationale for doing so in the manuscript text, or remove these references and replace them with relevant current references. Any changes to the reference list should be mentioned in the rebuttal letter that accompanies your revised manuscript. If you need to cite a retracted article, indicate the article’s retracted status in the References list and also include a citation and full reference for the retraction notice. 

Response: We have re-checked the reference list and ensured there is no retracted paper cited in our manuscript. There is no new reference included in the current revision.

Comment: Thanks for your efforts in editing the previous comments. Only the title of your manuscript should include the type of the study 

Response: We have modified the title by mentioning that this is a validity study of an intervention study protocol. Full title from ‘Consultation on the relevance and feasibility of a home hazard management program for falls prevention among community dwelling stroke survivors’ was changed to ‘A validity study to consult on a protocol of a home hazard management program for falls prevention among community dwelling stroke survivors’. Short title from ‘Feasibility of a home hazard management program for falls prevention in stroke’ was changed to ‘Validity study of a home hazard management program for falls prevention in stroke’. In consequence, we have corrected the sub-title in our main manuscript to align with the corrected title: ‘Relevance and feasibility’ was changed to ‘Protocol Validity’ (p.12).

---

## [Decision Letter · Decision Letter 2]

12 Dec 2022

A validity study to consult on a protocol of a home hazard management program for falls prevention among community dwelling stroke survivors

PONE-D-22-17675R2

Dear Dr. Romli,

We’re pleased to inform you that your manuscript has been judged scientifically suitable for publication and will be formally accepted for publication once it meets all outstanding technical requirements.

Kind regards,

Walid Kamal Abdelbasset, Ph.D.

Academic Editor

PLOS ONE

Additional Editor Comments (optional):

Reviewers' comments:

Reviewer's Responses to Questions

**Comments to the Author**

1. If the authors have adequately addressed your comments raised in a previous round of review and you feel that this manuscript is now acceptable for publication, you may indicate that here to bypass the “Comments to the Author” section, enter your conflict of interest statement in the “Confidential to Editor” section, and submit your "Accept" recommendation.

Reviewer #1: All comments have been addressed

Reviewer #2: All comments have been addressed

2. Is the manuscript technically sound, and do the data support the conclusions?

Reviewer #1: (No Response)

Reviewer #2: Yes

3. Has the statistical analysis been performed appropriately and rigorously? 

Reviewer #1: (No Response)

Reviewer #2: Yes

4. Have the authors made all data underlying the findings in their manuscript fully available?

Reviewer #1: (No Response)

Reviewer #2: Yes

5. Is the manuscript presented in an intelligible fashion and written in standard English?

Reviewer #1: (No Response)

Reviewer #2: Yes

6. Review Comments to the Author

Reviewer #1: (No Response)

Reviewer #2: The previously mentioned comments were done probably and a thankful efforts done by the author. The manuscript consumed a great effort

7. PLOS authors have the option to publish the peer review history of their article (what does this mean?). If published, this will include your full peer review and any attached files.

Reviewer #1: No

Reviewer #2: No

---

## [Editor Report · Acceptance letter]

2 Jan 2023

PONE-D-22-17675R2 

A validity study to consult on a protocol of a home hazard management program for falls prevention among community dwelling stroke survivors 

Dear Dr. Romli:

I'm pleased to inform you that your manuscript has been deemed suitable for publication in PLOS ONE. Congratulations! Your manuscript is now with our production department. 

Kind regards, 

on behalf of

Dr. Walid Kamal Abdelbasset 

Academic Editor

PLOS ONE